# Resilience of Lambs to Limited Water Availability without Compromising Their Production Performance

**DOI:** 10.3390/ani10091491

**Published:** 2020-08-24

**Authors:** Yusuf A. Adeniji, Musafau O. Sanni, Khalid A. Abdoun, Emad M. Samara, Mohamed A. Al-Badwi, Majdi A. Bahadi, Ibrahim A. Alhidary, Ahmed A. Al-Haidary

**Affiliations:** Department of Animal Production, College of Food and Agriculture Sciences, King Saud University, P.O. Box 2460, Riyadh 11451, Saudi Arabia; sfadeniji@gmail.com (Y.A.A.); mushafauoloyede@gmail.com (M.O.S.); dremas@ksu.edu.sa (E.M.S.); sh809090@gmail.com (M.A.A.-B.); vet.bahadi@gmail.com (M.A.B.); ialhidary@ksu.edu.sa (I.A.A.); ahaidary@ksu.edu.sa (A.A.A.-H.)

**Keywords:** growth, nutrient metabolism, Najdi sheep, water restriction

## Abstract

**Simple Summary:**

Feeding a pelleted diet under the prevailing water scarcity in arid regions, coupled with the low moisture content of that diet, raises the question about the precise level of water restriction that lambs can tolerate without compromising their production performance. Therefore, this study aimed to evaluate the production performance of lambs subjected to different levels of water restriction, which in the long run could help in the rationalization of the water consumption of the livestock production sector in arid and semi-arid regions. Lambs subjected to drinking-water restriction demonstrated efficient water use and conservation by drastically reducing water loss in feces and urine. Although dry-matter intake was decreased as a result of restricting water intake, the animals still gained reasonable body weight. It is surprising that the efficiency of nitrogen utilization was improved with the increasing level of water-intake restriction. The findings of the study revealed that lambs could tolerate up to 33% of water-intake restriction, depending on the climatic conditions and the type of diet.

**Abstract:**

Water scarcity is a common phenomenon in arid and semi-arid regions, which could have tremendous effects on livestock production. This study aimed to determine the level of water restriction that lambs fed on a pelleted diet can tolerate without compromising their production performance. A total of 24 male Najdi lambs were housed individually and randomly allocated into three equal groups, namely ad libitum water intake, 33% water-intake restriction, and 67% water-intake restriction. Dry-matter intake, feed conversion ratio, and average daily gain were decreased (*p* < 0.05) with the increasing level of water restriction. Water restriction had also reduced (*p* < 0.05) nutrient digestibility. The water-conserving ability of the water-restricted lambs was manifested by the production of concentrated and lower (*p* < 0.05) quantities of urine and feces. Meanwhile, serum osmolality and concentrations of albumin, total protein, urea-N, glucose, and non-esterified fatty acids were increased (*p* < 0.05) with the increasing levels of water restriction. It is surprising that lambs subjected to 67% water restriction retained more (*p* < 0.05) nitrogen relative to intake and had better (*p* < 0.05) efficiency of nitrogen utilization. It was strongly evident that lambs could tolerate water-intake restriction of up to 33% without compromising their production performance.

## 1. Introduction

Prolonged drought in arid and semi-arid regions results in limited availability of water sources [1]. Moreover, with global warming and climate change, both major and minor water bodies are beginning to dry up, resulting in water scarcity. Limitation in water intake and/or availability triggers in sheep certain physiological responses as a means of coping with this stress [2]. The relevance of water in ruminant production has been reviewed [3,4]. Management practices, such as overcrowding, could create competition among animals whereby young or weak animals are at a disadvantage and can face water-intake restriction [5,6]. Watering frequency is another management practice that could expose animals to water restriction. Animals that have access to water once or to a percentage of their ad libitum intake or that face water deprivation have increased thirst sensation, drink available water within 1–2 min, and stay without water until the next watering period [7]. Livestock native to arid regions have developed adaptive mechanisms to maintain high water economy by mobilizing their body reserves in order to compromise production performance even in periods of water scarcity [8]. However, several studies have succinctly shown that adequate water intake and roughage consumption are essential for optimal performance [9,10,11,12]. Under normal conditions (freely available feed and water), water intake and energy metabolism are related [13]. Thus, digestible energy is a sufficient benchmark in calculating water requirements for small ruminants. A 3:1 increase in daily water intake as the percentage increase of concentrate in the diet of sheep and goats fed salt-brush hay has been reported [14]. However, smaller water intake for goats and sheep fed diets with higher energy levels was observed [15]. It is interesting that lactating Aardi goats [16] and Awassi ewes [17,18,19] are able to withstand water restriction under high ambient temperature, although at the cost of production performance. Improved digestibility and efficient nutrient utilization have been previously reported as a consequence of the reduction in dry-matter intake (DMI), which allows rumen microflora to digest the consumed feed effectively in water-deprived animals [20,21,22].

Feeding a pelleted diet under the prevailing water scarcity in arid regions, coupled with the low moisture content of the pelleted diet, raised the question about the precise level of water restriction that lambs can tolerate without compromising their nutrient digestibility and production performance. Therefore, this study intended to evaluate the body physiology, nutrient metabolism, and production performance of lambs subjected to different levels of water restriction, which in the long run could help in the rationalization of water consumption of the livestock production sector in arid and semi-arid regions.

## 2. Materials and Methods

This study was conducted at the Experimental Station of the Animal Production Department, College of Food and Agriculture Sciences, King Saud University, Riyadh, Saudi Arabia. The Riyadh region is characterized by hot climatic conditions and extreme aridity with little rainfall throughout the year. The study was conducted at the end of the winter and beginning of the spring season where the mean ambient temperature (Ta), relative humidity (RH), and temperature-humidity index (THI) were 20.01 ± 0.52 °C, 49.33 ± 1.53%, and 64.61 ± 0.58, respectively. Ambient temperature (Ta) and relative humidity (RH) were continuously recorded every 30 min throughout the study period by using two data loggers (HOBO Pro-Series data logger, model H08-032-08, Onset Computer Corporation, Cape Cod, MA, USA) mounted about 2 m above the animals, and protected from heat, solar radiation, and water. A special data logging software (Box-Car Pro 4, Onset Computer Corporation) was used to program the loggers and for data retrieval. THI was thereafter calculated by using the following formula [23]: THI = Ta − (0.55 − 0.55 × RH) × (Ta − 58), where Ta is the ambient temperature in degrees Fahrenheit and RH is the relative humidity as a fraction of the unit. Daily meteorological data that prevailed during the study period are presented in Figure 1.

A total of 24 male Najdi lambs (four months old and 38 ± 0.24 kg body weight (BW)) were used in a 42-day study. Male lambs were used in this study because they are fast growers and more commonly used for meat production compared to females. Lambs were weighed, ear-tagged, vaccinated against clostridia diseases, and prophylactically treated for internal and external parasites. Thereafter, the animals were housed individually in shaded pens. All lambs were fed at 4% of their body weight on a commercial pelleted diet (Table 1) once daily at 08:00 h. The commercial pelleted diet consisted of alfalfa hay, barley, corn, wheat bran, soybean meal and crust, molasses, vitamins, and minerals. In addition, a block of mineral mixture was provided in each pen throughout the study period (Na 39 mg/kg, Mg 1000 mg/kg, Cu 350 mg/kg, Co 350 mg/kg, I 55 mg/kg, Mn 860 mg/kg, Zn 800 mg/kg, Fe 3950 mg/kg, and Se 40 mg/kg).

This study included a preliminary period (7 days) which was dedicated to the determination of the lambs’ average daily water intake. In fact, a known quantity of water (in excess of their requirements) was offered, and daily water intake was then calculated by subtracting the leftover after 24 h in the water trough from the water provided. The average daily water intake measured during this period was 4.5 ± 0.09 L per lamb, which was later used as a benchmark for specifying the different restriction treatment levels. The preliminary period was succeeded by a 42-day experimental period, at the beginning of which lambs were randomly assigned to one of three experimental treatment groups (eight lambs per treatment), namely 0% water restriction (W-0), 33% water restriction (W-33), and 67% water restriction (W-67) of the ad libitum average daily water intake. The amounts of water offered per treatment were adjusted weekly to correct for body weight change of these lambs where the ad libitum water intake for the control group (W-0) was measured weekly, and the restriction levels in groups W-33 and W-67 were adjusted accordingly.

For the sake of ethical considerations and animal welfare, the hydration level of the animals was tested weekly by using indices, such as skin turgor, hair coat, and eye clarity, as well as the shape and position of the eye within the orbit. In fact, all procedures adopted in this study were in accordance with Animal Welfare Act of Practice for the Care and Use of Animals for Scientific Purposes and approved by the Research Ethics Committee, King Saud University (KSU-SE-20-18).

Feed sampling was utilized at the beginning of the study and then weekly throughout the study period. These samples were stored frozen at −20 °C until the end of the study when they were pooled (5% of the total sample), ground, and analyzed for nutrient composition. DM was determined by drying the samples in a forced-air oven at 105 °C for 24 h while ash content was determined by burning the previously-dried samples at 550–600 °C in a muffle furnace for 3 h. Crude protein (CP) was produced following the Kjeldahl method, while neutral detergent fiber (NDF), acid detergent fiber (ADF), and ether extract (EE) were all produced as previously described [24,25].

On a daily basis, lambs were fed and given water at the same time (08:00 h), and the feed intake was monitored and recorded throughout the study period. To achieve this, the weight of the feed provided and the refusals were taken daily to calculate the daily DMI by using a sensitive scale that measured weights to the nearest 10 g. Meanwhile, lambs were weighed (BW) with an electronic scale (weight to the nearest 0.10 kg) prior to the morning feeding at the beginning of the experiment and then weekly throughout the experimental period. Weekly BW were taken so as to calculate the average daily gain (ADG) while the total weight gain (TWG) was computed as the difference between the final weights (FW) and initial BW. Feed conversion ratio (FCR) was thereafter calculated by dividing the daily DMI by ADG.

Ten mL of blood samples were collected from six lambs per treatment before the morning feeding via jugular venipuncture on days 0, 14, 28, and 42 into plain vacutainer tubes for serum separation. Serum samples were obtained by centrifuging the collected blood samples at 1600× *g* for 15 min at 4 °C, and then stored frozen at −20 °C until further analysis. Serum concentrations of glucose, total protein, albumin, non-esterified fatty acid (NEFA), urea nitrogen (urea-N), and creatinine were analyzed by using commercial kits (Randox Laboratories, Antrim, UK) and a semi-automated chemistry analyzer (RX Monza, Randox Laboratories Ltd., Crumlin, UK) according to the manufacturer’s procedures. Meanwhile, serum osmolality was measured by using an osmometer (VAPRO Pressure Osmometer, Model 5600, South Logan, UT, USA).

Rumen fluid samples were collected from six lambs in each treatment and analyzed for pH and osmolality. About 50 mL of rumen fluid sample was collected from each lamb by using an oral stomach tube fitted to a vacuum pump designed for the purpose. Samples were collected one hour before the morning feeding on day 0 and then every two weeks throughout the experimental period. After collection, rumen fluid was poured into glass tubes and labelled accordingly. The pH of rumen fluid samples was immediately analyzed by using a micro-electronic pH-meter (Model pH 211, Hanna Instruments, Woonsocket, RI, USA). Subsequently, 2 mL of rumen fluid samples were strained through four layers of cheesecloth, transferred into Eppendorf tubes, and stored frozen at –20 °C for later determination of the osmolality by using an osmometer (VAPRO Pressure Osmometer, Model 5600, South Logan, UT, USA).

At the end of the 42-day experimental period, five lambs from each treatment group were transferred to metabolic cages under the same water-intake restriction regime for a digestibility trial of seven days. The first four days served as an adaptation period, while data collection took place in the last three days of the trial. During the collection period, the daily DMI of each animal was recorded by subtracting the refusals from the amount of provided feed. In addition, feces weight (g) and urine volume (mL) of each animal were measured and recorded daily at 08:00 h. Both surfaces of urine collector and separator were coated daily with 100 mL of HCl in order to prevent nitrogen (N) volatilization. During the sampling process, urine pH was adjusted with HCl so that it remained below 3. A representative urine subsample (10%) was then taken, mixed thoroughly, and stored at −20 °C until further analysis. Similarly, representative subsamples of feed (5%) and feces (20%) were stored at −20 °C until further analysis. The representative subsamples of feed and feces were later pooled and analyzed for DM, CF, NDF, ADF, CP, EE, and ash contents [25]. Before analysis, samples were thawed and cooled at room temperature, air dried at 60 °C, ground, and passed through a 1-mm sieve. The DM content was determined by drying a known weight of the sample in a forced-air oven at 105 °C for 24 h. The new weight, as a proportion of the initial weight, signified the amount of DM as a percentage. Ash content was determined by the complete burning of samples in a muffled furnace at temperatures from 550–600 °C for 3 h.

All data generated were analyzed as a completely-randomized design using the PROC GLM procedure of SAS (SAS Institute Inc., Cary, NC, USA) to determine the differences in all parameters as a function of the fixed effect of the treatment (water-restriction levels: W-0, W-33, and W-67). The PROC MEANS procedure was used to obtain the descriptive statistics of all parameters. Finally, data were subjected to ANOVA, where means showing significant differences (*p* < 0.05) were tested by using the PDIFF option. The results were presented as mean ± SEM.

## 3. Results

Water restriction resulted in a significant (*p* < 0.05) decline in the DMI of lambs (Table 2). However, the DMI of lambs subjected to 33% restriction of drinking water (W-33) was almost similar to that of the control group (W-0) during the last two weeks of the experimental period (Figure 2). It was noticed that in less than 10 min and in a single bout, lambs in the restricted groups drank all the water provided for the W-67 group. However, the W-33 group drank the water in 2–3 bouts. Besides, the ADG and TWG were also affected (*p* < 0.05) by water restriction. In fact, low ADG (80 g/d) was recorded for lambs exposed to 67% water restriction while lambs exposed to 33% water restriction showed insignificant (*p* > 0.05) reduction in the ADG compared to the control group (170 g/d and 210 g/d, respectively) (Table 2). Meanwhile, the TWG of water-restricted lambs were 33% and 61% of the control for the W-67 and W-33 groups, respectively. Surprisingly, water-restricted lambs had gained a total of 3.08 kg and 5.67 kg body weight for the W-67 and W-33 groups, respectively, compared to a TWG of 9.3 kg for the control group, but the final BW was significantly (*p* < 0.05) lower in the 67% water-restricted group, albeit only numerically (*p* > 0.05) lower in the 33% water-restricted group compared to the control group. In addition, the FCR of the water-restricted lambs was significantly (*p* < 0.05) increased at the level of 67% water restriction, albeit only numerically (*p* > 0.05) increased at the level of 33% water restriction (Table 2).

The amount of feces and urine excreted, as well as the percentages of DM and N in the fecal materials of lambs in the different experimental groups are presented in Table 3. Restricting water intake had significantly (*p* < 0.05) reduced both fecal and urinary output. However, lambs exposed to 33% water restriction excreted the highest (*p* < 0.05) amount of feces compared to other groups. The lambs exposed to 67% water restriction excreted feces with the highest (*p* < 0.05) dry-matter content and N percent compared to the other groups. It has been observed that water loss in urine and feces was reduced with the increasing level of water restriction (Table 3).

The amount of nitrogen consumed and excreted in both feces and urine was altered (*p* < 0.05) in lambs exposed to water-intake restriction compared to the control group (Table 3). The amount of nitrogen consumed and excreted by lambs was not significantly (*p* > 0.05) affected by 33% water-intake restriction compared to the control group. However, the amount of nitrogen consumed and excreted by lambs was decreased (*p* < 0.05) at the level of 67% water restriction. This resulted in the reduction (*p* < 0.05) of the amount of nitrogen retained in the water-intake-restricted groups compared to the control group. However, the amount of retained N (as a percentage of N intake) and the percentage of retained N from absorbed N (RFA) were highest (*p* < 0.05) in lambs exposed to 67% water-intake restriction.

The mean digestibility (%) of the main nutrients is presented in Table 4. Digestibility of DM, CP, CF, NDF, ADF, and OM was higher (*p* < 0.05) for lambs provided with ad libitum water compared to those exposed to different levels of water-intake restriction whereas digestibility of EE was not (*p* > 0.05) affected by the water-intake restriction.

A reduction (*p* < 0.05) in the rumen pH was observed in the lamb group exposed to 67% water restriction compared to the other treatment groups (Table 5). However, water-intake restriction at the 33% level did not change (*p* > 0.05) the rumen pH compared to that of the control group. It is noteworthy that rumen fluid of the water-intake-restricted groups, during the collection period, was more viscous than that of the control group, and, in some cases, little rumen fluid was recovered, necessitating increasing the collected volume. Meanwhile, the feed particles in the rumen, at the end of the sixth week of the study, were coarser and drier in the water-intake-restricted groups compared to those of the control group. On the other hand, osmolality of the rumen fluid tended to increase (*p* = 0.052) in lambs exposed to both levels of water-intake restriction compared to those provided with ad libitum water intake (Table 5).

Regarding serum metabolites, concentrations of total protein, albumin, urea-N, glucose, NEFA, and osmolality were increased (*p* < 0.05) in water-restricted groups compared to those of the control group (Table 6). However, the concentration of serum creatinine shows a numerical (*p* > 0.05) increase with an increasing level of water-intake restriction.

## 4. Discussion

Water scarcity is a common phenomenon in arid and semi-arid regions of the world, which could have tremendous effects on livestock production. Shifting in the dynamics of body water balance, and thus multiple body responses, are considered major effects of water deprivation. This study was consequently conducted to determine the water restriction level that lambs could tolerate without compromising their body physiology, nutrient metabolism, and production performance. This could eventually help us rationalize the water consumption of the livestock production sector in arid and semi-arid regions. Water-deprivation and restriction studies on small ruminants have always shown a negative effect on the BW and feed intake as well as on the DMI during the restriction period. Variation between the studies depends on the level and duration of water restriction and on the prevailing ambient temperature. The average THI value prevailing throughout the study was 64.61, where it fluctuated between a minimum value of 57.42 and a maximum value of 70.87. For more than five decades, THI has been utilized to assess heat stress in farm animals by using the integrative effect of Ta and RH as a one-dimensional approach where values of 70 or less are generally considered comfortable [26]. Therefore, lambs used in this study were considered to be under comfortable climatic conditions throughout the study period.

Many studies on small ruminants have pinpointed the significant effects of different water-restriction regimens on production performance, particularly feed intake [27,28,29]. This study shows a decline in the DMI, ADG, and final BW of sheep with an increasing level of drinking-water restriction. These results are in agreement with those of previous studies on water-restricted Aardi does [16] where the reported level of decline in the DMI was proportional to that obtained herein although Alamer’s study was conducted under hot summer conditions. Similarly, a reduction in the DMI was reported in 50% water-restricted Baluchi lambs [30] and in two or four days, water deprived Awassi ewes [17]. It is interesting that lambs subjected to 33% water-intake restriction in this study, showed a DMI similar to that of the control group during the last two weeks of the study. This indicates that lambs might have developed some sort of adaptation to the low water-intake level. Similarly, German Fawn showed no difference in DMI when subjected to 87% and 73% restriction of their ad libitum water intake [31]. The reduction in DMI under water-restriction regimes could also be linked to the available type of feed [22]. In water-restriction experiments conducted on sheep, goats, and cattle, the effect of reduced feed intake is usually compensated for by increased feed-retention time with beneficial effects on digestibility and feed utilization [21,22].

A decreased BW gain was observed in lambs exposed to 67% water-intake restriction, while no effects were observed at 33% water-intake restriction. This finding supports a previous report on 33% water-restricted non-pregnant does that showed a BW gain similar to that of their control counterparts [32]. The observed reduction in BW gain at 67% water restriction could be a direct consequence of water-restriction-associated decrease in dietary intake [17,18,33]. The reduction in BW gain could be due to body water loss on one hand, and on the other hand, it could be attributed to the mobilization of fat used for energy metabolism to compensate for the decrease in DMI [17]. Similar to the findings reported herein, a steep decline in the BW of does [16] and ADG of lambs [30] was reported at 67% water-intake restriction. It is worth mentioning that similar to those reported in other studies, lambs in this study gained some weight despite the adopted water-intake restriction regime [29,34,35]. Such gain may be attributed to the prevailed Ta on one hand and the diet type (pelleted diet) on the other hand. Numerous studies have substantially shown that the processing methods of reducing the physical form of forages used in pelleted diet have effects on the rate at which digesta leave the rumen, which in turn controls voluntary feed intake, digestibility, and utilization [36,37,38].

A higher FCR was observed for lambs subjected to the highest level of water restriction (67%) compared to other treatments. This is mainly attributed to the observed higher reduction in the ADG compared to the decline in the daily DMI, as well as the reduction in nutrient digestibility due to water-intake restriction. Accordingly, lambs subjected to 67% water-intake restriction consumed twice as much feed to produce one-unit weight compared to the other treatment groups. It is interesting that lambs subjected to 33% water restriction had an FCR similar to that of the control group. This indicates that Najdi lambs could tolerate water-intake restriction of up to 33% without compromising nutrient utilization.

In this study, the volume of urine excreted and the amount of moisture in feces were decreased with the increasing level of water restriction. These observations are part of the physiological mechanisms that ruminants develop towards water stress as a result of the action of antidiuretic hormone and aldosterone on the renal and gastrointestinal tracts [2,12,39,40,41], and are in agreement with previous reports on lambs [42] and Awassi sheep [43].

On the other hand, lambs were in a positive N balance state, which may reflect the efficient N utilization [44,45]. This was reflected in the observed positive weight gain reported herein despite the adopted water-restriction regime. However, the water-restriction regime adopted in this study reduced the amount of N intake and consequently the amount of retained N. This indicates that the observed reduction in N retention in water-restricted groups is mainly attributed to the observed decline in the DMI rather than changes in the efficiency of N utilization. Water restriction also reduced the nutrient digestibility of the lambs with the exception of EE. These findings could be attributed to the known water-stress reduction of the number of rumen fauna [46], which may have contributed to the observed reduction in the nutrient digestibility of lambs subjected to water restriction. Moreover, a slight reduction in the rumen liquor pH and elevation in the rumen liquor osmolality were observed for lambs subjected to water restriction compared to those provided with water ad libitum. The observed reduction in the rumen pH could be attributed to the reduced saliva secretions and increased volatile fatty acids (VFA) production in water-restricted lambs [30] while the observed increase in rumen osmolality could be attributed to the net water flow across the rumen epithelium and the increased rumen Na^+^ level in dehydrated animals [41,47]. These effects are consistent with previous reports in goats, cows, and camels [41,48,49]. Nevertheless, rumen pH and osmolality values recorded throughout the study were within the normal physiological range [50,51,52].

The level of serum osmolality and several metabolites monitored herein were higher in the lambs subjected to water restriction. The observed rise in serum osmolality could be attributed to the decrease in blood volume as a result of the restriction of drinking water. Although plasma/serum volume was not measured during the study, previous studies have shown the relevance of serum osmolality in determining the dehydration status of small ruminants [29,43]. The observed higher serum concentration of total proteins in water-restricted lambs might help in maintaining the blood oncotic pressure in water-stressed animals as the loss of water results in hemoconcentration and hypovolemia [53]. It is interesting that the observed lower DMI herein due to water restriction did not result in a decreased serum concentration of total proteins, as previously reported in ewes [18,54]. Rather, the observed increase in the serum concentrations of total protein and albumin in water-restricted lambs could be attributed to the hypovolemic status of the animals [55]. In fact, the serum albumin level was not affected by 33% water-intake restriction, which indicates that Najdi lambs could tolerate water restriction of up to 33% without showing any signs of dehydration. The increase in serum concentrations of urea observed in this study could also be attributed to the hypovolemic status of the animals [55] as well as the herein-observed low dietary protein intake in the drinking-water-restricted lambs. However, the values reported in the present study are within the normal physiological range for sheep [56,57], which does not indicate any kidney malfunction. Similarly, serum concentrations of glucose and NEFA were also increased in lambs subjected to water restriction; however, their values remained within the normal physiological range reported for sheep [56,58]. Therefore, it could be inferred that even with the decline in DMI, the lambs’ nutrient requirement was not grossly impacted. On the other hand, the level of serum creatinine was not affected by the different levels of water restriction adopted in the current study. This might indicate that the rate of muscle proteolysis was low or rather that there was no pressure on the muscles for extra sources of energy, even with the reduced DMI in water-restricted lambs [59].

## 5. Conclusions

Male Najdi lambs subjected to drinking-water restriction demonstrated efficient water use and conservation by drastically reducing water loss in feces and urine. Although DMI was decreased as a result of restricting water intake, the animals still gained reasonable body weight. It is surprising that the efficiency of N utilization was improved with the increasing level of water-intake restriction. The results obtained indicate that male Najdi lambs could tolerate water-intake restriction of up to 33%, depending on the climatic conditions and the type of diet. Further research is definitely imperative to determine the level of water restriction tolerable, without compromising the body metabolism and production performance, under hot environmental conditions.

## Figures and Tables

**Figure 1 animals-10-01491-f001:**
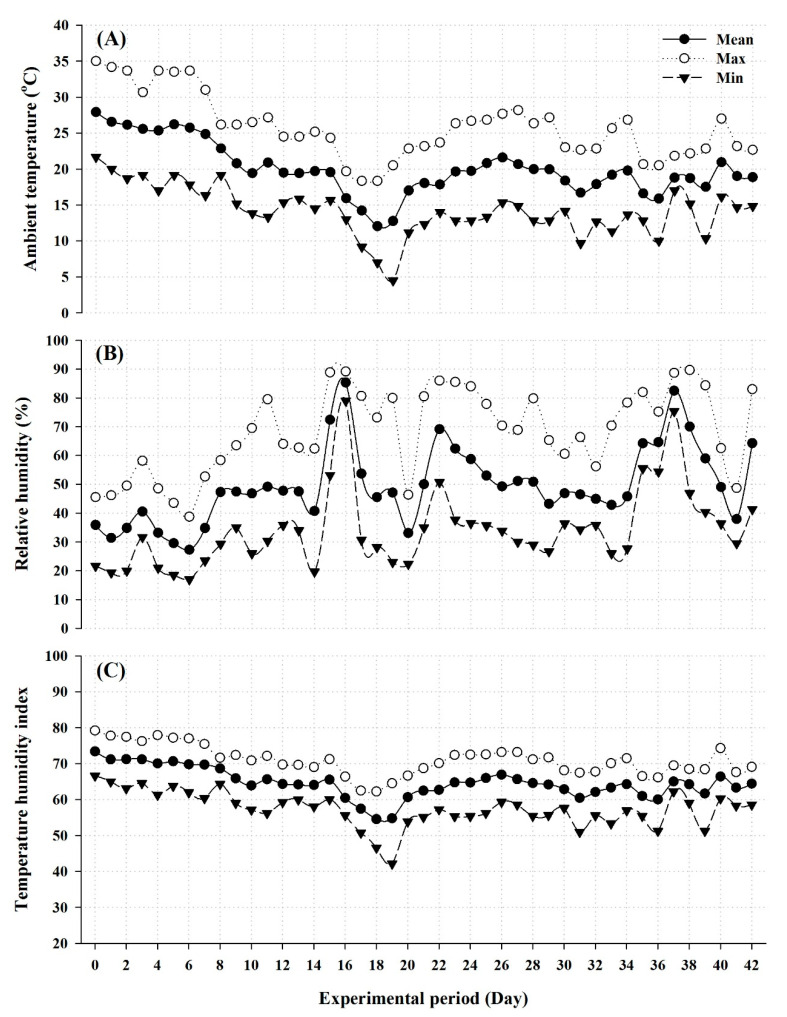
Daily maximum, minimum, and mean ambient temperature (**A**), relative humidity (**B**), and temperature humidity index (**C**) throughout the experimental period.

**Figure 2 animals-10-01491-f002:**
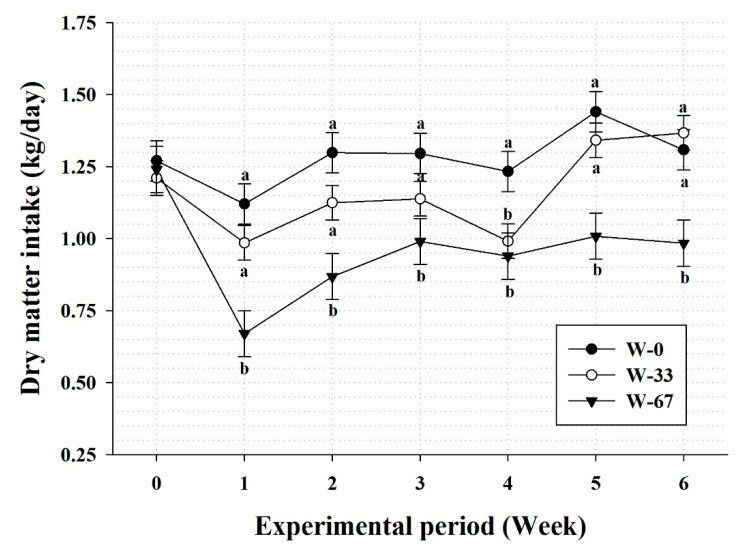
Dry-matter intake of water-restricted male Najdi lambs fed a pelleted diet throughout the experimental period. W-0: ad libitum water intake, W-33: 33% restriction of ad libitum water intake, and W-67: 67% restriction of ad libitum water intake. ^ab^ Measurements within the same week with different letters are significantly different at *p* < 0.05.

**Table 1 animals-10-01491-t001:** Nutrient (proximate) composition of the pelleted diet fed to male Najdi lambs during the study.

Nutrients	Composition (DM Basis)
Dry matter (DM, %)	92.43
Ash (%)	7.79
Organic matter (OM, %)	92.21
Crude protein (CP, %)	13.04
Ether extract (EE, %)	3.70
Neutral detergent fiber (NDF, %)	49.28
Acid detergent fiber (ADF, %)	23.45
Metabolizable energy (MJ/kg) ^1^	11.30

^1^ Metabolizable energy is calculated and MJ: megajoule.

**Table 2 animals-10-01491-t002:** Production performance of water-restricted male Najdi lambs fed a pelleted diet.

Parameters ^1^	Treatments ^2^			SEM	*p* Value
	W-0	W-33	W-67		
Duration (days)	42	42	42	-	-
Initial BW (kg)	38.75	38.84	38.90	0.24	0.675
Final BW (kg)	48.05 ^a^	44.51 ^a^	41.98 ^b^	1.20	0.006
DMI (kg/d)	1.28 ^a^	1.16 ^b^	0.91 ^c^	0.03	<0.001
ADG (kg/d)	0.21 ^a^	0.17 ^a^	0.08 ^b^	0.02	0.003
TWG (kg)	9.30 ^a^	5.67 ^b^	3.08 ^c^	0.16	0.015
FCR (DMI: gain)	6.10 ^a^	6.82 ^ab^	11.38 ^b^	0.64	0.026

^abc^ Means within the same row with different letters are significantly different at *p* < 0.05. ^1^ DMI: dry-matter intake, BW: body weight, ADG: average daily gain, TWG: total weight gain, and FCR: feed conversion ratio. ^2^ W-0: ad libitum water intake, W-33: 33% restriction of ad libitum water intake, and W-67: 67% restriction of ad libitum water intake.

**Table 3 animals-10-01491-t003:** Nitrogen metabolism in water-restricted male Najdi lambs fed a pelleted diet.

Parameters ^1^	Treatments ^2^			SEM	*p* Value
	W-0	W-33	W-67		
N intake (g/d)	26.52 ^a^	25.74 ^a^	22.19 ^b^	1.01	0.024
Fecal output (g/d)	632 ^b^	738 ^a^	554 ^c^	0.01	<0.001
Fecal DM (%)	49.05 ^b^	46.61 ^b^	55.60 ^a^	0.01	0.001
N fecal (g/d)	9.31 ^ab^	10.01 ^a^	8.55 ^b^	0.30	0.015
N fecal (%)	1.47 ^ab^	1.36 ^a^	1.54 ^b^	0.27	0.039
N absorbed (g/d)	17.01 ^a^	15.74 ^b^	13.65 ^c^	0.18	<0.001
N absorption (%)	64.63 ^a^	61.59 ^b^	61.46 ^b^	0.16	<0.001
Urine output (mL/d)	1046.6 ^a^	338.6 ^b^	327.1 ^b^	0.05	<0.001
N urine (g/d)	3.44 ^a^	4.22 ^a^	2.08 ^b^	0.57	0.037
N excreted (g/d)	12.75 ^a^	14.23 ^a^	10.63 ^b^	0.87	0.035
N retained (g/d)	13.78 ^a^	11.51 ^b^	11.56 ^b^	0.30	<0.001
N retention (%)	51.66 ^a^	45.03 ^b^	51.79 ^a^	1.02	<0.001
N RFA (%)	81.01 ^b^	73.12 ^c^	84.69 ^a^	0.15	<0.001

^abc^ Means within the same row with different letters are significantly different at *p* < 0.05. ^1^ DM: dry matter, N: nitrogen, RFA: retained from absorbed = (retained/absorbed) × 100. ^2^ W-0: water intake, W-33: 33% restriction of ad libitum water intake, and W-67: 67% restriction of ad libitum water intake.

**Table 4 animals-10-01491-t004:** Apparent digestibility (%) of nutrients in water-restricted male Najdi lambs fed a pelleted diet.

Parameters ^1^	Treatments ^2^			SEM	*p* Value
	W-0	W-33	W-67		
DM	75.46 ^a^	72.19 ^b^	70.78 ^b^	0.84	0.006
CP	64.63 ^a^	61.59 ^b^	61.46 ^b^	0.16	<0.001
EE	93.09	92.67	93.28	1.44	0.954
CF	58.89 ^a^	48.67 ^c^	54.75 ^b^	0.92	<0.001
NDF	77.55 ^a^	74.04 ^b^	72.59 ^b^	0.78	0.002
ADF	60.63 ^a^	51.66 ^b^	50.94 ^b^	0.75	<0.001
OM	78.21 ^a^	73.64 ^b^	74.08 ^b^	0.66	0.001

^abc^ Means within the same row with different letters are significantly different at *p* < 0.05. ^1^ DM: dry matter, CP: crude protein, EE: ether extract, CF: crude fiber, NDF: neutral detergent fiber, ADF: acid detergent fiber, and OM: organic matter. ^2^ W-0: ad libitum water intake, W-33: 33% restriction of ad libitum water intake, and W-67: 67% restriction of ad libitum water intake.

**Table 5 animals-10-01491-t005:** Rumen liquor pH and osmolality in water-restricted male Najdi lambs fed a pelleted diet.

Parameters	Treatments ^1^			SEM	*p* Value
	W-0	W-33	W-67		
pH	6.66 ^a^	6.61 ^a^	6.43 ^b^	0.07	0.046
Osmolality (mosm/L)	253	269	277	12.50	0.052

^ab^ Means within the same row with different letters are significantly different at *p* < 0.05. ^1^ W-0: ad libitum water intake, W-33: 33% restriction of ad libitum water intake, and W-67: 67% restriction of ad libitum water intake.

**Table 6 animals-10-01491-t006:** Serum metabolites in water-restricted male Najdi lambs fed a pelleted diet.

Parameters ^1^	Treatments ^2^			SEM	*p* Value
	W-0	W-33	W-67		
Albumin (g/dL)	3.39 ^b^	3.47 ^b^	3.80 ^a^	0.04	0.025
Total protein (g/dL)	5.65 ^c^	6.60 ^b^	7.30 ^a^	0.06	0.013
Urea N (mg/dL)	26.29 ^c^	34.22 ^b^	55.82 ^a^	0.52	0.001
Glucose (mg/dL)	82.60 ^b^	92.41 ^a^	96.15 ^a^	2.15	0.005
NEFA (mmol/L)	0.50 ^c^	0.57 ^b^	0.64 ^a^	0.57	0.028
Osmolality (mosm/L)	279.77 ^b^	307.25 ^a^	317.61 ^a^	5.49	0.021
Creatinine (mg/dL)	1.63	1.76	1.77	0.13	0.584

^abc^ Means within the same row with different letters are significantly different at *p* < 0.05. ^1^ N: nitrogen, and NEFA: non-esterified fatty acids. ^2^ W-0: ad libitum water intake, W-33: 33% restriction of ad libitum water intake, and W-67: 67% restriction of ad libitum water intake.

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
