# Peer review of "Resilience of Lambs to Limited Water Availability without Compromising Their Production Performance"

_animals, 2020, doi:10.3390/ani10091491_

Round 1

Reviewer 1 Report

Dear authors,

your work is interesting and actual.

I have some comments:

in general, Latin words have to be written in italics (et al.; ad libitum). Please correct in whole manuscript (for example: L50; 100; 193; 194; 208; 209; 227; 228; 241; 242; 250; 251; 479; 480....).

L78; 117; 119; 120; 151; 153; 155; 157: please write: °C    not oC!!!

L83: what is BW? Please explain abbreviation when it first appears.

L86: I would write the sentence as follows: All lambs were fed at 4% of their body weight on commercially pelleted total mixed ration (Table 1) once daily at 08:00 h.

Table 1: Mcal is not SI unit. You must recalculate it!

L147: add space -> 100 ml

Table 2 - table 6: letters in tables write as upper superscripts!

Sincerely,

reviewer.

Reviewer 2 Report

The manuscript describes the effect of water deprivation on growth performance in lambs. Although limiting water should always be avoided under practical conditions, this manuscript confirm the data available in the literature showing a negative effect of decreasing water intake. The study has good control and a reasonable number of experimental units giving it a good statistical power to discriminate treatment effect.

Overall, the manuscript is well written, and I have only few suggestions regarding the language.

General Comments and Questions:

  1. In my understanding you fed the animals a pelleted diet not a TMR. So, I suggest replacing “total mixed ration pellets” and “TMR pellets” for “pelleted diet” throughout the manuscript. Please consider doing that.
  2. The authors monitored the meteorological measurements (L. 108-115, and Figure 1) but they did not mention if it was out of sheep comfort temperature. Please consider adding this at the beginning of the discussion section.
  3. Crude fiber has such a limited application in ruminant nutrition that it doesn’t worth even show it. So, I suggest removing CF, because it is not adding to the discussion.
  4. The authors should remove FI since you are ready reporting DMI.

Specific Comments:

Line 20. I suggest replacing the word “appreciable”

Line 23. Do you want to say “depending” instead of “pending”?

Line 64. This sentence is a bit confuse. Decreasing DMI usual increase ruminal retention time and further digestibility. As it is put here, it seems that the animal will “effectively” digest the feed if it is water restricted. Please, rephrase this sentence.

Table 1. Is the Metabolizable energy calculated or measured? Please add a footnote.

Line 96. Please add a SD for the measured water intake.

Line 118. Were these samples ground?

Line 120. This time is different than that shown on L. 157. Is it right?

Line 124. I suggest replacing “In daily basis” by “In a daily basis”

Line 133. I suggest adding “of” after “mL”

Results. Did the authors measure water intake throughout the study? If positive, you should show it on table 2.

Table 2. I think the parameters are presented here out of the conventional other. I suggest: Initial BW, final BW, DMI, water intake, ADG, TWG, and FCR. Duration can be removed since it is the same for all treatments.

Table 2. ADG and DMI units are wrong, it should be (Kg/day)

Table 2. I suggest replacing the very low p-values “0.000” by “<0.001”. Please do the same for all tables.

Line 195. I suggest replacing “voided” by “excreted” or another similar word.

Table 3. Please double-check all parameters units.

Table 3. As in table 2, the parameters are presented out of an expected order. Please rearrange it.

Table 3. The data of retained N is very intriguing. How did the W-67 lambs show similar retained N than W-33 but a way lower ADG? How can we explain that?

Table 3. Another intriguing data is the absorbed N in percentage, which I think is the same as digestibility. Why the absorbed nitrogen is significantly different among treatments, but the CP digestibility is not? Please elaborate on this in the discussion section.

Table 4. Please add the “apparent digestibility” to the table title.

Line 276. Please check this sentence.

Line 277. Do you mean “ruminal microflora”?

Line 284-285. I don’t think the W-67 lambs were mobilizing muscle. They had a nitrogen retention of 11.5 g/d.

Line 306. “lambs”

Reviewer 3 Report

Dear Authors,

attached please find a pdf with comments.

Regards

Reviewer 4 Report

I would like to review the statistical analysis and the presentation of the results in order to make it more understandable and get more out of the information generated.

My suggestions are indicated in the draft, with the corresponding comments.

Round 2

Reviewer 3 Report

Dear Authors,

accept in present form.

Regards

Author Response

Thanks for accepting the article in its present form.